# Fabrication and Characterisation of the Cytotoxic and Antibacterial Properties of Chitosan-Cerium Oxide Porous Scaffolds

**DOI:** 10.3390/antibiotics12061004

**Published:** 2023-06-03

**Authors:** Lemiha Yildizbakan, Neelam Iqbal, Payal Ganguly, Eric Kumi-Barimah, Thuy Do, Elena Jones, Peter V. Giannoudis, Animesh Jha

**Affiliations:** 1School of Chemical and Process Engineering, University of Leeds, Leeds LS2 9JT, UK; 2Leeds Institute of Rheumatic and Musculoskeletal Medicine, University of Leeds, Leeds LS9 7JT, UK; 3Division of Oral Biology, School of Dentistry, University of Leeds, Leeds LS9 7TF, UK; 4Academic Department of Trauma and Orthopaedic Surgery, School of Medicine, University of Leeds, Leeds LS2 9JT, UK

**Keywords:** cerium oxide, chitosan, antibacterial scaffold, cytotoxicity, tissue engineering

## Abstract

Bone damage arising from fractures or trauma frequently results in infection, impeding the healing process and leading to complications. To overcome this challenge, we engineered highly porous chitosan scaffolds (S1, S2, and S3) by incorporating 30 (wt)% iron-doped dicalcium phosphate dihydrate (Fe-DCPD) minerals and different concentrations of cerium oxide nanoparticles (CeO_2_) (10 (wt)%, 20 (wt)%, and 30 (wt)%) using the lyophilisation technique. The scaffolds were specifically designed for the controlled release of antibacterial agents and were systematically characterised by utilising Raman spectroscopy, X-ray diffraction, scanning electron microscopy, and energy-dispersive X-ray spectroscopy methodologies. Alterations in the physicochemical properties, encompassing pore size, swelling behaviour, degradation kinetics, and antibacterial characteristics, were observed with the escalating CeO_2_ concentrations. Scaffold cytotoxicity and its impact on human bone marrow mesenchymal stem cell (BM-MSCs) proliferation were assessed employing the 2,3-bis-(2-methoxy-4-nitro-5-sulfophenyl)-2H-tetrazolium-5-carboxanilide (XTT) assay. The synthesised scaffolds represent a promising approach for addressing complications associated with bone damage by fostering tissue regeneration and mitigating infection risks. All scaffold variants exhibited inhibitory effects on bacterial growth against *Staphylococcus aureus* and *Escherichia coli* strains. The scaffolds manifested negligible cytotoxic effects while enhancing antibacterial properties, indicating their potential for reducing infection risks in the context of bone injuries.

## 1. Introduction

Bone tissue engineering is a burgeoning multidisciplinary field which aims to develop advanced synthetic or natural bone scaffolds that effectively promote the regeneration of damaged or diseased bone tissues. Open fractures represent a substantial medical concern, given that the exposed bone is highly susceptible to contamination and subsequent infections [1]. Infections associated with open fractures in the upper or lower extremities can precipitate a myriad of complications, including protracted fracture union, impaired wound healing, development of chronic osteomyelitis, extended hospitalisation, escalated utilisation of antibiotics, and compromised quality of life [1]. A comprehensive management strategy is indispensable for treating open fractures, encompassing wound debridement, fracture stabilisation, and prophylactic measures against infections [2].

Conventional bone substitutes comprise allografts, autografts, and xenografts, which entail the utilisation of bone fragments procured from the patient’s own body, a human donor body, or a non-human animal body, respectively [3]. Biological grafts pose considerable risks, such as disease transmission, immunogenic reactions, and graft rejection, despite their potential efficacy. Furthermore, their availability is inherently limited, and their applicability or effectiveness may be inadequate [4]. In light of these limitations, researchers are exploring synthetic biomaterials as alternative therapeutic options. To foster bone regeneration while mitigating the risk of infection, synthetic bone scaffolds must fulfil a range of criteria. The scaffold’s pore size, morphology, and interconnectivity are critical determinants of cellular and tissue responses in the scaffold–tissue interface. An optimal bone scaffold should exhibit a porous architecture, facilitating the transport of nutrients and metabolic waste products and providing mechanical stability to the surrounding tissue [5]. Lyophilisation, commonly referred to as freeze-drying, is a prevalent technique for producing porous scaffolds, owing to its capacity to conserve the scaffold material’s physical and chemical attributes while facilitating control over pore size and shape, as well as preserving the biological activity of the integrated bioactive agents [6]. The versatility of freeze-drying, which enables its application across various materials, renders it a suitable option for bone scaffold fabrication. Extensive investigations were conducted on porous bone scaffolds, owing to their propensity to create a supportive milieu for cellular attachment, proliferation, and differentiation, thereby catalysing new bone tissue formation.

Bone scaffolds with inherent antibacterial properties are of paramount importance in bone tissue engineering, as they facilitate successful bone regeneration [7] while concurrently inhibiting bacterial infections [6]. In recent years, cerium oxide nanoparticles (CeO_2_) have emerged as a promising candidate in this domain, owing to their unique antibacterial characteristics and biocompatibility with mammalian cells. The antimicrobial activity of CeO_2_ nanoparticles can be attributed to two primary mechanisms: the reversible redox transition between Ce^3+^ and Ce^4+^ oxidation states [8] and the generation of reactive oxygen species (ROS). The latter are potent oxidising agents capable of inflicting substantial damage to bacterial cell membranes, ultimately compromising their structural integrity and functionality [9]. Incorporating CeO_2_ nanoparticles into bone scaffolds yields a biocompatible and antibacterial composite material, effectively mitigating the risk of bacterial infections that may otherwise hinder the bone healing process. 

Adsorption, oxidoreduction, and toxicity were suggested as three possible interactions between CeO_2_ nanoparticles and bacteria [10]. The first mechanism involves CeO_2_ nanoparticles, which carry a positive charge, attaching to negatively charged bacterial cell walls through electrostatic interactions. This attachment likely blocks the membrane and lingers, impairing the cell wall’s viscosity and disrupting the transport exchange between the solution and bacterial cells. Oxidoreduction, the second kind of interaction, happens close to the bacterial wall. High cytotoxicity is caused by oxidoreduction, which is connected to oxidative stress in bacteria during nanoparticle adsorption. The key to understanding how CeO_2_ nanoparticles affect the survival of bacterial cells is oxidative stress. The fact that the toxicity persisted after 1 or 5 h of contact suggests the existence of a quick mechanism for the nanoparticles’ transmembrane adsorption altering the bacterial membrane, corrupting particular ionic pumps, and, as a result, significantly altering the interaction of the cell with the solution and decreases cell viability [11]. The third mechanism involves generating reactive oxygen species (ROS) that induce oxidative stress on the bacterial cell wall surfaces due to the reversible conversion of Ce^3+^ and Ce^4+^. ROS are known to attack nucleic acids, proteins, and polysaccharides, resulting in loss of function and eventually leading to the destruction and decomposition of bacteria [8].

Tissue engineering has increasingly recognised the potential of chitosan as a biomaterial due to its biodegradable nature, biocompatibility with living tissues, ability to promote osteoblast cell proliferation and attachment [12], and inherent antibacterial properties. The antimicrobial properties of chitosan can be ascribed to its capacity to interact with and disrupt bacterial cell membranes, thereby impairing their function [13]. A recent study reported that chitosan-based porous scaffolds demonstrated antibacterial properties against *S. aureus* and *E. coli* while promoting osteogenic differentiation of human bone marrow mesenchymal stem cells (hBMSCs) [14]. However, chitosan alone cannot fully emulate the properties of natural bone. Consequently, composite materials combining chitosan and calcium phosphates (CaPs) are often employed to address this limitation [6]. Despite their utility, CaPs such as hydroxyapatite (HA), dicalcium phosphate dihydrate (DCPD), and octacalcium phosphate (OCP) minerals exhibit brittleness, which constrains their capacity to support mechanical loads following the stabilisation of extensive bone defects. To circumvent this limitation, researchers have discovered that doping CaPs with iron (Fe) can enhance the materials’ osteogenic potential.

Fe ions were shown to enhance the presence of the protein necessary for cell adhesion compared to undoped DCPD samples [15]. As a result, the existence of Fe ions in DCPD may positively impact cell proliferation behaviour [16]. Furthermore, since iron is a naturally occurring element in the human body, its incorporation into the scaffold poses minimal risk of toxicity or rejection as Fe^3+^ is crucial for the blood’s haemoglobin carrying oxygen [15]. Additionally, Fe doping of CaPs has been shown to improve toughness and durability while promoting the growth of new bone tissue by augmenting osteogenic potential, expediting the healing process, and reducing the likelihood of complications [17]. Therefore, Fe-doped CaPs in bone tissue engineering represent a promising avenue for future research and development, with the potential to significantly improve patient outcomes and advance the field of bone tissue engineering applications.

Synthetic bone scaffolds have the potential to revolutionise the treatment of open fractures by providing a safe, effective, and easily obtainable solution for bone regeneration. They can offer several advantages over traditional bone substitutes, including the ability to be customised to meet specific clinical needs and incorporate antibacterial properties. This study aimed to fabricate and characterise highly porous chitosan scaffolds embedded with iron-doped dicalcium phosphate dihydrate minerals (Fe-DCPD) and cerium oxide nanoparticles to investigate bone tissue engineering applications’ osteogenic and antibacterial properties.

## 2. Results

### 2.1. Characterisation Results

#### 2.1.1. Raman Spectroscopy

The Raman spectra of freeze-dried chitosan and a combination of Fe-DCPD mineral, CeO_2_, and chitosan scaffolds are shown in Figure 1. Chitosan, derived from chitin, has similar Raman peaks such as the C-O-C stretching vibration at 890 cm^−1^, C-C stretching and C-O stretching vibrations at 1060 cm^−1^, CH_2_ deformation at 1370 cm^−1^, CH_3_ deformation at 1420 cm^−1^, and Amide I peak at 1650 cm^−1^. The prominent Raman peaks for these calcium phosphates are the ν1 symmetric stretching of PO_4_^3−^ at 960 cm^−1^, ν3 asymmetric stretching of PO_4_^3−^ at 1046 cm^−1^, ν4 bending mode of PO_4_^3−^ at 590 cm^−1^, and ν2 bending mode of PO_4_^3−^ at ~430 cm^−1^ [17]. The exact peak positions and intensities may vary based on the iron doping level and specific calcium phosphate phase. Cerium oxide is a well-known rare-earth oxide with a fluorite structure and is part of the analysed scaffolds. The dominant Raman peak for cerium oxide is the symmetric stretching mode of the Ce-O bond at 456 cm^−1^, observed in all samples containing cerium oxide nanoparticles but not in the sample only containing chitosan. It is worth noting that particle size, crystal structure, impurities, or other phases can influence peak positions and intensities. A detailed table of Raman spectra peak assignments is provided in Table 1, which outlines peak positions and corresponding assignments for all the observed peaks in the Raman spectra of the tested scaffolds. The analysis of the Raman spectra is vital in understanding the chemical compositions and structures of the examined scaffolds, aiding in comprehending their properties and applications in diverse fields such as tissue engineering and regenerative medicine.

#### 2.1.2. X-ray Diffraction

The results of experimental XRD diffraction patterns for the CeO_2_, CH, S1, S2, and S3 are illustrated in Figure 2. The peaks from the obtained DCPD pattern match the reference standard XRD data for CeO_2_ (JCPDS 00-067-0123) compiled by the Joint Committee on Powder Diffraction and Standards (JCPDS). Two broad 2 peaks at 10° and 20° correspond to the crystal I and II phase forms, respectively, forming the distinctive XRD fingerprint of the partly crystalline polysaccharide CS [23]. The CH diffraction pattern shows two broad peaks at 9.2° and 20.2° for the freeze-dried CH scaffold, which indicates excellent CS purity due to the absence of other peaks. Due to the high crystallinity of CeO_2_ nanoparticles, it was expected that increasing CeO_2_ concentration would decrease the CS crystal II phase form for the freeze-dried scaffolds containing CeO_2_.

The results (Figure 2) indicate that the crystallite size and percentage of the crystallinity of the scaffolds increased with an increase in the concentration of cerium oxide, as demonstrated in Figure 2c. XRD results revealed the crystalline structure of the CeO_2_ nanoparticles, CH, S1, S2, and S3 scaffolds are illustrated in Figure 2a. Strong peaks were observed for all samples (CeO_2_, S1, S2, and S3) at 28.2°, 32.7°, 47.2°, 56.0°, 58.8°, 69.0°, 76.5°, and 78.89° corresponding to the (111), (200), (220), (311), (222), (400), (331), and (420) crystalline planes. As the CeO_2_ concentration increases, the partially crystalline CS structure becomes more crystalline, and the S3 diffraction pattern becomes similar to the CeO_2_ pattern, as shown in Figure 2a. Figure 2b displays the XRD diffraction pattern for the fluorite CeO_2_ JCPDS 00-067-0123. The crystalline sizes were determined via the Debye Scherrer equation, where CeO_2_, S1, S2, and S3 were 7.9, 6.1, 6.8, and 6.4 nm, respectively (Figure 2c). The percentage crystallinities for CH, CeO_2_, S1, S2, and S3 were calculated to be 0.11%, 88%, 82%, 80%, and 80.3%, respectively. The results indicate that increasing the cerium oxide concentration reduced the crystalline size while increasing the crystallinity.

#### 2.1.3. Scanning Electron Microscopy and Energy Dispersive X-ray Spectroscopy (EDX)

The incorporation of CeO_2_ resulted in changes to the morphology and structural features of the scaffolds, affecting the pore size distribution and lamellae thickness while increasing the size of the pores. When the concentration of CeO_2_ in the freeze-dried scaffold was increased to 30 (wt)%, the shape of the pores became less defined. Figure 3 displays the morphological characterisation and porous size distribution of the freeze-dried samples containing varying concentrations of CeO_2_. The SEM images indicate that all scaffolds have a well-connected and multilayer pore structure, with an open pore size of around 60 µm determined by ImageJ analysis software (https://imagej.nih.gov/ij/, accessed on 23 December 2022). The pore size of S1 scaffolds ranges from 0 to 160 µm, S2 ranges from 10 to 120 µm, and S3 ranges from 0 to 140 µm.

#### 2.1.4. Ultra-Violet Visible Spectroscopy

Figure 4 illustrates the absorption spectrum obtained from the commercially available (Sigma Aldrich, Darmstadt, Germany) and synthesised CeO_2_ nanoparticles dispersed in deionised water for comparison. The UV-visible absorption spectrum reveals two distinct broad bands occurring at 207 nm (peak at A) and 303 nm (peak occurs at B) in both samples. The absorption peak observed at 207 nm corresponds to the characteristic peak of the Ce^3+^ ion, while 303 nm is ascribed to the Ce^4+^ ion [24]. The CeO_2_ absorption spectrum and peaks correlate with CeO_2_ nanoparticles absorption synthesised using the precipitation method from cerium nitrate as precursor solution, which is reported elsewhere [24]. The broad absorption bands observed in the UV region between 250 nm to 400 nm may favour charge transfer transitions from the O 2p to Ce 4f energy levels (O (2p)→Ce (4f)) instead of the Ce 4f-4f spin-orbit splitting of the Ce 4f states [25]. In comparison, the presence of Ce^3+^ at 207 nm enables the excitation of an electron from the 4f to the 5d shell.

UV-vis absorption spectra aimed to evaluate the coexistence of Ce^3+^ and Ce^4+^ ions, known as biological antioxidants and antibacterial agents, as demonstrated in Figure 4. We showed that the synthesised CeO_2_ nanoparticles exhibit a higher Ce^3+^ to Ce^4+^ ratio of 0.96 compared to 0.89 of the commercially available nanoparticles, indicating that the synthesised CeO_2_ nanoparticles exhibit better antibacterial properties.

Following the CeO_2_ nanoparticle suspension UV-visible absorbance analysis, two experiments comprising *E. coli* dispersed in deionised water and *E. coli* suspension in 1 mL synthesised CeO_2_ solution as discussed in Figure 4A were investigated for their UV-visible spectral features variability. The comparison between the UV-visible spectra of *E. coli* and *E. coli* suspension in CeO_2_ solutions is shown in Figure 4B, with the inset clearly showing the prominent absorption bands. The *E. coli* absorption coefficient spectral reveals three dominant absorption bands centred at 203 nm (dotted red line), 260 nm (peak C), and 330 nm (peak E). Conjugating *E. coli* bacteria in CeO_2_ solution, the spectral absorption coefficient shows the same three prominent absorption peaks obtained from *E. coli* with an additional peak at 307 nm (peak D). The broadband absorption observed from 295 nm to 350 nm represents an overlap between *E. coli* and Ce^4+^ ion. The deep UV absorption coefficient of *E. coli* bacteria in CeO_2_ solution at 203 nm drops significantly by a factor of four (not fully shown). The absorption in the wavelength ranging from 230 nm to 700 nm increases upon dispersing *E. coli* into CeO_2_ solution. It is essential to mention that by dispersing *E. coli* in the CeO_2_ nanoparticle-suspended solution, the absorption band of the Ce^4+^ ion in the deep UV is completely suppressed while that of Ce^3+^ is enhanced.

Different absorption spectra were obtained by subtracting the CeO_2_ absorption spectrum from the *E. coli* suspension in CeO_2_ nanoparticle spectral, as shown in Figure 5b. The differential absorption results significantly drop compared to the *E. coli* spectrum indicating that *E. coli* is suppressed.

Similarly, Figure 6a represents the UV-visible absorption spectra of the *S. aureus* bacterial and *S. aureus* suspension in CeO_2_ solution with the inset showing. Before dispersing *S. aureus* bacteria into CeO_2_ suspension solution, the *S. aureus* shows strong absorption peaks at 215 nm, 266 nm, and 323 nm. After dispersing the *S. aureus* bacteria in the CeO_2_, similar absorption peaks were observed; however, the deep UV absorption became more intense and shifted to the lower wavelength. The absorption spectrum ranges from 250 nm to 700 nm enhanced, suppressing Ce^3+^ and Ce^4+^ ions absorption bands. Moreover, Figure 6b illustrates a comparison between *S. aureus* and the differential absorption spectra of *S. aureus*–CeO_2_. The drop in absorption spectra shows that the *S. aureus*–CeO_2_ combination reduces the *S. aureus* bacterial activity, similar to the *E. coli*–CeO_2_ mixture discussed above.

### 2.2. Testing Results

#### Swelling and Degradation

The swelling results for fabricated freeze-dried scaffolds containing different amounts of cerium oxide nanoparticles (10, 20, and 30 (wt)%) are shown in Figure 7a. It is observed that, in the 0 to 30 min time range, all the scaffolds present a fast-swelling increment. However, the increment speed for the 30 to 150 min range drastically diminished, and it reached mass stability between 150 and 270 min. The results indicate that increasing the cerium oxide concentrations caused an increase in the scaffold swelling percentages. Scaffolds S1, S2, and S3 presented swelling percentages of 506.9 ± 27.92%, 440 ± 19.08, and 385.6 ± 20.3%, respectively. The freeze-dried sample degradation results are presented in Figure 7b. Increasing the cerium oxide concentration reduced the mass loss of the scaffolds, with S3 showing the lowest mass loss of 7.9 ± 1.01%; however, 10 (wt)% cerium oxide scaffolds’ mass loss was 18.3 ± 1.76% after 4 weeks.

### 2.3. In Vitro Cell Results

#### 2.3.1. Direct Cytotoxicity

Images of control well and wells with MSCs and scaffolds were captured on days 1, 3, and 7. Images from day 7 are shown in Figure 8. After the completion of day 7, there was no change in the colour of the media, and neither was any turbidity observed, indicating that the cells were healthy. Additionally, the morphology of the cells at the cell-scaffold interface also did not change and remained comparable to the control; thus, confirming that the scaffolds were not cytotoxic to the BM-MSCs.

#### 2.3.2. Indirect Cytotoxicity

Indirect cytotoxicity was performed in two parts, one was indirect cytotoxicity, and the other was proliferation by XTT. The results of the indirect extract cytotoxicity experiment (Figure 9a) show over 90% viability of BM-MSCs when exposed to extracts collected over all the time points, namely 3, 7, and 14 days. The extract collected from scaffolds was non-toxic and indicated high viability compared to the positive control. While no statistical differences were observed between the % live cells exposed to the extracts of each scaffold, the data indicate that all three scaffold samples supported the BM-MSCs.

Next, the proliferative activity of the BM-MSCs was investigated using the extracts collected for up to 14 days; this was carried out by exposing the BM-MSCs to the extracts for up to 5 days to allow them to proliferate with positive and negative controls. The experiment used 250, 500, and 1000 cells/well cell concentrations. There was no significant difference in the trend observed, and data are presented for 500 cells/well (Figure 9b). Cell proliferation is comparable to positive controls for all freeze-dried scaffolds, demonstrating that CeO_2_ do not inhibit cell proliferation and growth.

### 2.4. Antibacterial Test Results

Antibacterial properties were examined in CeO_2_ nanoparticles, chitosan, and the synthesised scaffold types (S1, S2, and S3) against Gram-positive (*S. aureus*) and Gram-negative (*E. coli*) bacteria.

The growth of bacteria in the CeO_2_, chitosan, and the synthesised scaffolds with different concentrations of the CeO_2_ was analysed after 24 h to see release agents of the CeO_2_. The results are shown in Figure 10. Control bacteria (CB) represents the control bacteria growth experiment, chosen as a reference for comparison with the growth experiments in the absence and presence of cerium oxide scaffold samples. 

Antibacterial properties were examined in all three scaffolds against Gram-positive (*S. aureus*) and Gram-negative (*E. coli*) bacteria. Since chitosan is also known to exhibit antibacterial properties, it was essential to use this as a control to compare with the CeO_2_ nanoparticles and scaffolds containing varying concentrations of CeO_2_ nanoparticles. CB is the control for bacterial growth without the addition of the samples. Results demonstrated in Figure 10 indicate that the scaffolds reduced bacterial proliferation. *S. aureus* reduced bacterial growth by 41 ± 14.4%, 61.5 ± 9.8, 76 ± 15%, 82 ± 11%, and 88 ± 6% for chitosan, CeO_2_, S1, S2 and S3 scaffolds, respectively. In contrast, *E. coli* bacterial growth for chitosan, CeO_2_, S1, S2, and S3 scaffolds was reduced by 55 ± 10%, 63.5 ± 11%, 68 ± 12%, 79 ± 8%, and 84 ± 7%. Based on the results, all scaffold types express antibacterial properties compared to CB. The scaffolds containing increasing concentrations of CeO_2_ nanoparticles presented increased antibacterial efficacy compared to the control scaffolds (freeze-dried chitosan only). Table 2 summarises the results of the synthesised scaffolds.

## 3. Materials and Methods

### 3.1. Cancellous Synthetic Bone

#### 3.1.1. Chitosan Solution

An amount of 3 (wt)% chitosan (molecular weight (M_w_)) of 3100 to 3750 kDa and degree of deacetylation (DD) ≥ 75%) was prepared in a 2 % (*v*/*v*) acetic acid solution. The solution was stirred with a magnetic stirrer for 24 h. After the stipulated time, the beaker was covered with aluminium foil and left undisturbed overnight to allow air bubbles to rise to the solution surface. The chitosan solution was stored at 4 °C and was utilised to fabricate a synthetic cancellous bone scaffold.

#### 3.1.2. Iron-Doped Brushite (Fe-DCPD)

A 0.1 M aqueous solution (200 mL) of Ca(NO_3_)_2_∙4H_2_O (Fisher Chemicals, CAS:13477-34-4) was heated to 37 °C and designated as solution A. A 0.1 M solution (200 mL) of (NH_4_)_3_PO_4_ (Acros Organics, CAS:7783-28-0) was mixed with 10 (mol)% iron nitrate powder (Fe(NO_3_)_3_·9H_2_O) (VWR Chemicals, CAS:7782-61-8) and added dropwise to solution A while continuously stirring at 37 °C for 2 h. The mixture was left to settle for 1 h, allowing the precipitation of the Fe-DCPD (CaHPO_4_·2H_2_O). The precipitated crystals were then collected on filter paper (Whatman grade 44 with 3 μm pores), washed multiple times with distilled water, and dried for 24 h at 80 °C.

#### 3.1.3. Cerium Oxide Nanoparticles (CeO_2_)

The nanoparticles were synthesised using a hydroxide-mediated method, employing cerium nitrate hexahydrate (Ce(NO_3_)_3_·6H_2_O, Sigma-Aldrich, CAS:10294-41-4) as a precursor. In brief, 10.85 g of Ce(NO_3_)_3_·6H_2_O(s) was dissolved in 250 mL of distilled water and stirred continuously for 20 min, yielding a 0.1 M solution (A). Next, 0.3 M sodium hydroxide (NaOH, Sigma-Aldrich, CAS: 1310-73-2) solution was added dropwise to the solution (A) at 50 °C under continuous magnetic mixing to facilitate the hydrolysis of cerium oxide nanoparticles. The solution was covered with aluminium foil and maintained at 50 °C under constant stirring for 24 h. The nanoparticles were filtered and washed five times with distilled water and ethanol. The recovered nanoparticles were frozen at −80 °C for 24 h and then subjected to freeze-drying at −100 °C and a pressure of 43 mTorr for 24 h. The synthesis reaction is represented in Equations (1)–(3) [8].
(1)$$\mathrm{Ce}{\left({\mathrm{NO}}_{3}\right)}_{3}6H_{2}O+3\mathrm{NaOH}\;\to \mathrm{Ce}{\left(\mathrm{OH}\right)}_{3}+3\mathrm{Na}({\mathrm{NO}}_{3})+6H_{2}O$$

Precipitation:(2)$${\mathrm{Ce}}^{3+}+3{\mathrm{OH}}^{-}\to \mathrm{Ce}{\left(\mathrm{OH}\right)}_{3\;\left(s\right)}$$

Oxidation:(3)$$4{\mathrm{Ce}}^{3+}+12{\mathrm{OH}}^{-}+O_{2}\to 4{\mathrm{CeO}}_{2\;\left(s\right)}+6H_{2}O$$

#### 3.1.4. Synthetic Cancellous Bone Scaffold

The cancellous region of the synthetic bone scaffolds was created by mixing chitosan, iron-doped brushite, and varying quantities of cerium oxide nanoparticles. The composition percentage of each component is tabulated in Table 3. The scaffolds were produced using a 10 mL suspension batch and stirred for 2 h on a hot plate to achieve a uniform mixture. The mixed suspensions were injected into well plates (24-well) and subsequently frozen at −80 °C for 24 h, then placed in a freeze-drier operating at 43 mTorr at −100 °C for 24 h.

### 3.2. Characterisation Techniques

The fabricated samples were analysed for molecular chemical characterisation with Renishaw inVia Raman spectroscopy at a 785 nm wavelength and 24.9 mW operating power. The laser beam was focused onto the sample’s surface using an ×50 microscope objective, and the frequency of the vibrational range was from 0 to 3000 cm^−1^.

X-ray diffraction, a non-destructive analytical method, analysed the samples. All the synthesised samples were subjected to X-ray powder diffraction to determine their crystalline structure using a D8 X-ray diffractometer with Cu K radiation (Kα = 0.15406 nm). The samples were scanned from the 10° to 80° Bragg angle 2θ range, with a 5 s scan time and a 0.03° step size. The recorded patterns were analysed using HighScore Plus software (https://www.malvernpanalytical.com/en/products/category/software/x-ray-diffraction-software/highscore-with-plus-option, accessed on 23 December 2022), and the Rietveld refinement method was used to evaluate the mineral samples’ crystallinity based on peak shape and intensity analysis. The X-ray diffraction analysis was conducted to determine the crystallite size and % crystallinity of the fabricated scaffolds (S1, S2, S3) and CeO_2_. The crystallite size was calculated using the Debye-Scherrer equation:D = 0.9λ/β cos(θ)(4)
where D represents the crystallite size (nm), λ is the wavelength, θ is the Bragg half angle (2θ), and the Bragg reflection full width at half-maximum (FWHM) is β (in radians). The % crystallinity of the fabricated scaffolds was evaluated by subtracting the area of crystalline peaks from the total area of all peaks.

The Hitachi SU8230 1–30 kV cold field emission gun scanning electron microscopy (SEM) was used to analyse the microstructure and determine the pore size and porosity of the freeze-dried scaffolds. Prior to SEM, the samples were treated with 6 µm of iridium to increase the materials’ electrical conductivity, which improved the signal-to-noise ratio.

The UV-Visble absorption spectrum of homogeneous clear CeO_2_ nanoparticles suspension in deionised water was initially measured using PerkinElmer LAMBDA 950 UV/VIS/NIR Spectrometer (PerkinElmer, Inc, Waltham, MA, USA) in the wavelength ranging from 200 nm to 800 nm. Before measuring each sample’s spectrum, the background spectrum was recorded using suspended media employed in sample preparation, which is deionised water. The CeO_2_ samples were prepared by mixing 5 mg CeO_2_ nanoparticles in 25 mL distilled water. The samples containing the bacterial strains (*S. aureus* and *E. coli*) and the CeO_2_ nanoparticles were prepared by adding 0.25 mL of the bacterial broth solutions with 25 mL of distilled water (solution A), then 1 mL of solution A was used to resuspend the CeO_2_ (suspended solution). All the absorbance measurements were carried out at room temperature with a 1 cm pathlength cuvette.

### 3.3. Testing Methods

#### 3.3.1. Swelling

Before testing, all freeze-dried samples were immersed in a 1 M NaOH solution for 5 min and washed twice with distilled water. The samples were dried at 60 °C for 24 h and weighed before the experiment. The swelling test was performed using Dulbecco’s phosphate-buffered saline solution (DPBS) (Life Technologies, Paisley, UK). The solution was poured into individual Eppendorf tubes, and the scaffold samples were immersed for 30 min at 37 °C. After removing the samples from the PBS solution, they were weighed again using an electronic balance. The percentage of sample swelling (*n* = 3) was determined using the following equation:(5)$$Swelling\;\left(\%\right)=\frac{Ww-Wd}{Wd}\times 100$$

Ww represents the wet weight, and Wd represents the dry weight of the samples. The process was repeated for up to 270 min.

#### 3.3.2. Degradation

The freeze-dried scaffolds were immersed in PBS solutions at 37 °C once a week; the samples were removed from the solution, dried for 24 h at 60 °C, and weighed. The samples were then immersed in fresh PBS solution, which was repeated for four weeks. The percentage degradation of the samples was calculated using the following equation:(6)$$\Delta Wd\;\left(\%\right)=\frac{Wo-Wd1}{Wd1}\times 100\;$$

Wo represents the initial sample weight, and Wd1 denotes the sample weight at time (t).

### 3.4. In Vitro Studies

In vitro, studies were performed with fabricated freeze-dried samples (S1, S2, S3) with a diameter of 1 cm and a height of 0.5 cm. Before commencing the in vitro investigations, the samples were cleaned using 70 (*v*/*v*)% ethanol and washed thrice with DPBS. Following this, the samples were sterilised by exposing each side to UV treatment for 60 min.

#### 3.4.1. Ethical Approval and Cell Culture

Ethical approval for the collection of samples was obtained from NREC Yorkshire and Humberside National Research Ethics Committee (number 06/Q1206/127). Bone marrow mesenchymal stem cells (BM-MSCs) were obtained from three healthy donors after informed written consent and processed to obtain mononuclear cells that were culture expanded to isolate BM-MSCs as previously described [26]. The cells expressed the MSC phenotype of CD105, CD73, and CD90 and were negative for CD45 [27]. Once confluent, the cells were frozen using 10% Dimethylsulfoxide (DMSO) (Thermo Scientific, Loughborough, UK), in 45% Dulbecco’s Modified Eagle Medium (DMEM) (Life Technologies, Paisley, UK) and 45% foetal bovine serum (FBS), (Thermo Scientific, Loughborough, UK) for future experiments.

Prior to experiments, frozen vials from *n* = 3 donors were defrosted, pooled, and placed into culture, and utilised at passage 3 (p3) for in vitro investigations. The procedure involved defrosting the frozen cells in a water bath at 37 °C and adding them to DMEM, supplemented with 10% FBS and 1% Penicillin/Streptomycin (P/S) antibiotics (both from Sigma, Dorset, UK). The cell suspension was centrifuged at 300× *g* and resuspended in complete MSC StemMACS media (SM) (Miltenyi Biotec, Bisley, UK). The cells were then placed in tissue culture flasks (T25) (Corning, New York, NY, USA) at the seeding density of 2 × 10^5^ in an incubator at 37 °C and 5% CO_2_ until nearly confluent and ready to be used. Half media changes were performed twice a week to maintain the cultures. Cells were detached for further use; the flasks were washed with DPBS and then treated with 5 mL Trypsin/ethylene diaminetetra acetic acid (EDTA) (both from Sigma, Poole, UK) incubated at 37 °C for 5–7 min. After this, 15 mL of DMEM with 10% FBS was added to the flask to stop the action of trypsin. The total cell suspension volume of 20 mL was centrifuged at 300× *g* to obtain a cell pellet. The cells were resuspended in complete DMEM media and counted using trypan blue.

#### 3.4.2. Direct Cytotoxicity

The direct toxicity assay was conducted according to the seven-day ISO10993-5:2009 protocol to investigate the direct impact of scaffolds on BM-MSCs. Sterilised scaffold samples (*n* = 3) were added to a 6-well plate and secured using steri-strips pieces (3 M steri strips cat no. R1540C, Medisave, UK). An amount of 5 × 10^4^ BM-MSCs was then added to each well in 2 mL SM media, and a control group consisting only of cells without scaffolds with SM was also included. Microscopic imaging of the interface between the scaffold and cells was carried out at 24 h, 72 h, and 7 days. Imaging was performed using pooled donor samples for each type of scaffold for up to 7 days. As per the guidelines of direct cytotoxicity testing, it is required to keep the materials to be tested for in the same media conditions for up to 7 days. Thus, there was no media change performed to adhere to the ISO protocol.

#### 3.4.3. Indirect Cytotoxicity

The indirect toxicity test aimed to detect any harmful effects of scaffold extracts on MSCs. Scaffold extracts were prepared by collecting 330 μL of media exposed to the scaffolds in 6 Eppendorf for each scaffold. The test conditions included a positive control (consisting only of SM), a negative control (10% DMSO in SM), and extracts from each scaffold in duplicate. The steps were performed as per the manufacturer’s protocol.

##### Cytotoxicity Assay

Three BM-MSC cultures (*n* = 3) were pooled, and the resulting cells were seeded in triplicate in 200 µL of SM media at 1 × 10^4^ cells/well in a 96-well plate for 24 h. The media was then removed and replaced with 100 µL of extracts (defrosted) containing either the scaffold eluate, negative control, or positive control for another 24 h before adding XTT reagents, as described below.

##### Proliferation Assay

BM-MSCs (*n* = 3) were pooled and seeded in a 96-well plate at densities ranging from 250 to 1000 cells per well in SM and incubated for 24 h. After the incubation period, the media was replaced with treatment media containing either the scaffold eluate, the negative control, or the positive control. The cells were then cultured for four days to assess cell proliferation by XTT. For XTT cell proliferation assay experiments, 5 mL of XTT labelling reagent (Sigma, Dorset, UK) was mixed with 0.1 mL of electron coupling reagent for one microplate (96 wells). Following exposure to scaffold extract, positive or negative control media, the wells were replaced with 100 µL of DMEM with 10% FBS and 50 µL of the XTT solution and incubated at 37 °C for 4 h. Subsequently, 100 µL of each well’s aliquot was transferred to the corresponding well of a new plate, which was read on a microplate reader (Cytation 5, Biotek) at 450 nm and 630 nm (reference wavelength). The optical density (OD) was calculated by subtracting the value for the reference wavelength at 630 nm from 450 nm. The ODs of the test wells were normalised to the ODs of the positive control to measure cell viability or proliferation inhibition.

### 3.5. Bacterial Cultures and Experiments

The antibacterial property of the scaffolds was investigated through viable counting and optical density measurements. The scaffold samples were washed once with 70 (*v*/*v*)% ethanol and thrice with Dulbecco’s phosphate-buffered saline (DPBS) solution. The samples were then sterilised in a furnace at 80 °C for 4 h. Each biofilm experiment was conducted in triplicates. Two bacterial strains, namely *S. aureus* and *E. coli*, were selected to examine the antibacterial properties of the scaffolds. *S. aureus* and *E. coli* bacteria were selected as they commonly cause post-orthopaedic surgical infections [28,29,30,31]. The bacterial strains were obtained from a −80 °C stock and were provided by the Oral Biology division at the University of Leeds’ School of Dentistry. The bacterial strains were streaked onto Brain Heart Infusion (BHI) agar plates. After 24 h of incubation at 37 °C, a single colony was selected from each type of bacteria and grown in 25 mL of BHI broth in an anaerobic cabinet at 37 °C for 24 h. This process allowed for the creation of fresh bacterial suspensions to be used for inoculation. Bacterial cultures in BHI broth were subjected to optical density measurements at OD600 nm using the Jenway 6305 UV/Visible Spectrophotometer. Bacterial suspensions incubated for 24 h at OD600 nm of 0.15 were utilised to ensure reproducibility. One hundred µL of bacterial suspensions and 900 µL BHI solutions were added to each 24-well plate. The sterilised scaffolds (CH, S1, S2, and S3) were transferred into the bacterial suspension with sterile tweezers. The plates were then incubated for 24 h at 37 °C in an anaerobic cabinet (Don Whitley Scientific). Biofilms from the scaffolds were resuspended in 1 mL PBS solution. Viable counting was carried out through serial dilutions (10^4^ times) in BHI and plating to estimate the bacterial abundance from each scaffold sample. After 24 h of incubation in the anaerobic cabinet, bacterial colonies were counted, and the colony-forming units (CFU)/mL were calculated.

For in vitro work, data were analysed using Graph Pad Prism (version 9.5.0). The data were grouped and analysed using Two-way ANOVA with Geiser greenhouse correction with matched values stacked across a row in the datasheet. Multiple comparisons were also investigated to compare the % live cells for each type of scaffold at every time point.

### 3.6. Statistical Analysis

For in vitro work, data were analysed using Graph Pad Prism (version 9.5.0). The data were grouped and analysed using two-way ANOVA with Geiser greenhouse correction with matched values stacked across a row in the datasheet. Multiple comparisons were also investigated to compare the % live cells for each type of scaffold at every time point. In addition, two-way ANOVA was performed for bacterial work to compare the antibacterial activity of each formulation S1, S2 and S3 against control for Gram-positive and Gram-negative bacteria.

## 4. Discussion

Bone scaffolds require porosity and interconnected pores to support bone cell adhesion, growth, differentiation, nutrient transport, and waste removal [32]. In the tissue engineering field, the standard pore size for these scaffolds ranges from 50 µm to 1500 µm [33]. Both small and large pores contribute to bone growth and blood vessel formation. As a result, scaffolds featuring multi-scale porosity can enhance vascularisation by promoting bone growth and blood vessel formation through various pore sizes, leading to improved scaffold vascularisation [6].

The bone scaffolds created using the freeze-drying technique displayed highly interconnected porous structures with diverse pore size distributions, as evidenced by SEM analysis (Figure 3). Smaller pore size distributions increase the scaffold surface area, providing more attachment points for cells. Large pores exceeding 1500 µm diminish the scaffold area, which has been reported to reduce cell attachment [34]. Pore diameters below 50 µm can hinder cell migration, create cellular capsules, and lead to necrotic zones in extreme cases due to limited nutrient and waste transport [35,36].

The scaffolds’ chemical structure was revealed through Raman spectroscopy. Raman spectroscopy has been employed to compare the characteristic bands in the freeze-dried scaffolds. The CeO_2_ [19], iron-doped brushite [17] and chitosan [18] peaks observed are similar to the literature. After four weeks, the S3 freeze-dried scaffolds experienced the lowest mass loss (7.9 ± 1.01%), while S1 scaffolds showed the highest mass loss (18.3 ± 1.76%). This difference is likely due to S3’s higher crystallinity than S1 and S2 scaffolds, as XRD analysis indicates. Other researchers have corroborated that mass loss is related to deacetylation (DD) level, molecular weight (M_w_), and crystallinity [37].

The synthesised scaffold degradation results indicate that increasing the CeO_2_ concentration reduced the scaffold mass loss. The freeze-dried S1 scaffolds demonstrated the most critical mass loss of 18.3 ± 1.76%, while the S3 scaffolds showed the lowest mass loss at 7.9 ± 1.01% after four weeks. The difference is likely attributed to the S3 exhibiting increased crystallinity, as verified from the XRD analysis compared to the other scaffolds (S1, S2). Increased crystallinity leads to extensive hydrogen bonding and intermolecular forces between the chitosan biopolymer chains resulting in a more compact scaffold structure, therefore decreasing the water molecule’s accessibility to the groups of hydrophilic.

A scaffold’s ability to retain water is crucial for determining its appropriateness for tissue engineering. Chitosan’s structure includes free amine groups, making it a hydrophilic polymer with high water absorption capacity [38]. The swelling properties of scaffolds were shown to significantly impact cell adhesion, proliferation, and differentiation. As a result, S1 scaffolds containing 10 (wt)% CeO_2_ exhibited the highest liquid absorption, while S3 scaffolds with 30 (wt)% CeO_2_ demonstrated the lowest swelling percentage increase. Mutlu et al. (2022) reported that as CeO_2_ concentration increased in chitosan scaffolds, the swelling ratio of the samples decreased [39], which also aligns with the results of our study.

Consequently, the pore size must be sufficiently large to enable cell migration throughout the scaffold while allowing cell adhesion [5,40]. The presence of CeO_2_ nanoparticles in the scaffolds impacted the pore size and porosity distribution. As CeO_2_ nanoparticle concentrations increased, the number of pores rose significantly, and pore size distribution decreased, ranging from 0 to 160 µm (S1), 10 to 120 µm (S2), and 0 to 140 µm (S3). Indirect cytotoxicity assay results (Figure 6) indicate that the porous size and porosity distribution of S1, S2, and S3 are suitable for cell growth. An increase in CeO_2_ concentration correlated with increased cell proliferation (Figure 7). Moreover, when calcium phosphate minerals are dispersed across the scaffold surface, more bone cells may interact with one another [41]. The freeze-dried bone scaffolds exhibit morphological features and highly interconnected porous structures with diverse pore size distributions, as verified by SEM analysis (Figure 3). The composite scaffolds displayed a rougher surface as cerium concentration (10 to 30 (wt)%) increased.

The antibacterial effectiveness of cerium oxide nanoparticles against the two bacterial strains was demonstrated for freeze-dried samples containing various CeO_2_ concentrations. The coexistence of the two oxidation states (Ce^3+^ and Ce^4+^) confirmed for UV-Vis analysis enabled the nanoparticles to express antibacterial behaviour through the ability to cycle between cerous (Ce^3+^) and ceric (Ce^4+^) via oxidation state-induced oxygen vacancies and reactive oxygen species [42]. In the presence of microbial activity, the pH reduces, which causes more protons to be produced; thus, electrons are released into the medium, contributing to the Ce^3+^ and Ce^4+^ ratio and characteristics.

Similar work was performed by Li et al. (2018) [43]; however, Alpaslan et al. (2015) found higher cytotoxicity levels concerning the CeO_2_ nanoparticles, potentially due to their higher concentrations [44]. It is possible to develop and apply biomaterials doped with Ce^3+^ and Ce^4+^ nanoparticles, which could lead to new methods for preventing and treating bone infections in high-risk patients, including those with diabetes, weakened immune systems, and in vulnerable areas that are susceptible to infection, such as avascular bone, open fractures, necrosis, and prolonged reconstruction procedures [44]. This strategy could also contribute to combating antibiotic resistance and the increasing prevalence of bone infections [8].

## 5. Limitations and Conclusions

The XRD, Raman, and SEM characterisation outcomes validated the successful synthesis of porous lyophilised chitosan scaffolds incorporating 30 (wt)% IB and diverse concentrations of CeO_2_ nanoparticles (10, 20, and 30 (wt)%. Upon increasing the CeO_2_ nanoparticle concentration from 0 to 30 (wt)%, scaffold crystallinity was enhanced, leading to a decrease in the degradation rate when submerged in 37 °C PBS. The elevated crystallinity at higher CeO_2_ concentrations contributed additional hydrogen bonding and intermolecular forces, constraining the chitosan biopolymer chains and reducing the total liquid absorption of the lyophilised scaffold. Pore size distributions contracted with an increase in CeO_2_ nanoparticle concentration (10 to 30 (wt)%). The antibacterial assessment indicated that escalating the CeO_2_ nanoparticle concentration from 10 to 30 (wt)% in the lyophilised scaffolds amplified their antibacterial capabilities. The scaffolds demonstrated potent antibacterial properties against both Gram-positive and Gram-negative bacterial strains. Although all scaffold types expressed antibacterial properties, the S3 variation will be further investigated due to presenting the highest antibacterial efficacy compared to S1 and S2 variations (Table 3).

Despite substantial advancements in the field of synthetic bone scaffolds, there remain challenges to overcome. These include optimising the scaffold’s mechanical properties to resemble natural bone closely and devising advanced techniques for scaffold delivery to the fracture site. In conclusion, applying antibacterial scaffolds in bone tissue engineering represents a promising approach for mitigating infections subsequent to bone damage. The highly porous chitosan scaffolds embedded with iron-doped brushite minerals and cerium oxide nanoparticles exemplify progress in this domain, demonstrating exceptional biocompatibility, mechanical integrity, and antibacterial properties. Further research is warranted to evaluate their in vivo efficacy and investigate potential clinical applications.

## Figures and Tables

**Figure 1 antibiotics-12-01004-f001:**
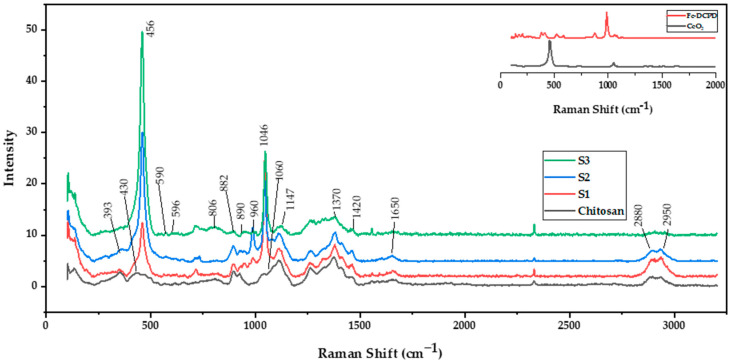
Renishaw inVia Raman spectrometer operating at 785 nm was used to characterise the molecular bonds of the chitosan freeze-dried scaffolds containing 10 (wt)% iron-doped brushite minerals containing different concentrations of CeO_2_, S1: 10 (wt)%, S2: 20 (wt)%, and S3: 30 (wt)%.

**Figure 2 antibiotics-12-01004-f002:**
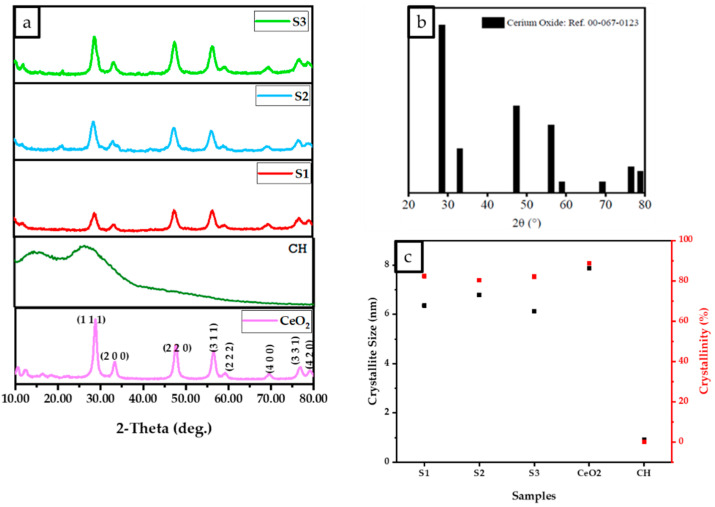
D8 X-ray diffractometer with Cu K radiation (Kα = 0.15406 nm), 10° to 80° Bragg angle as 2θ range, 5 s scan time and a 0.03° step size parameter were used to analyse the results of the normalised data. (**a**) The XRD data of freeze-dried chitosan, scaffolds, and cerium oxide (CeO_2_), (**b**) XRD pattern reference for CeO_2_ (JCPDS 00-067-0123 reference) and (**c**) The relationship between crystallinity and crystallite size for freeze-dried scaffolds (CH, S1, S2, and S3).

**Figure 3 antibiotics-12-01004-f003:**
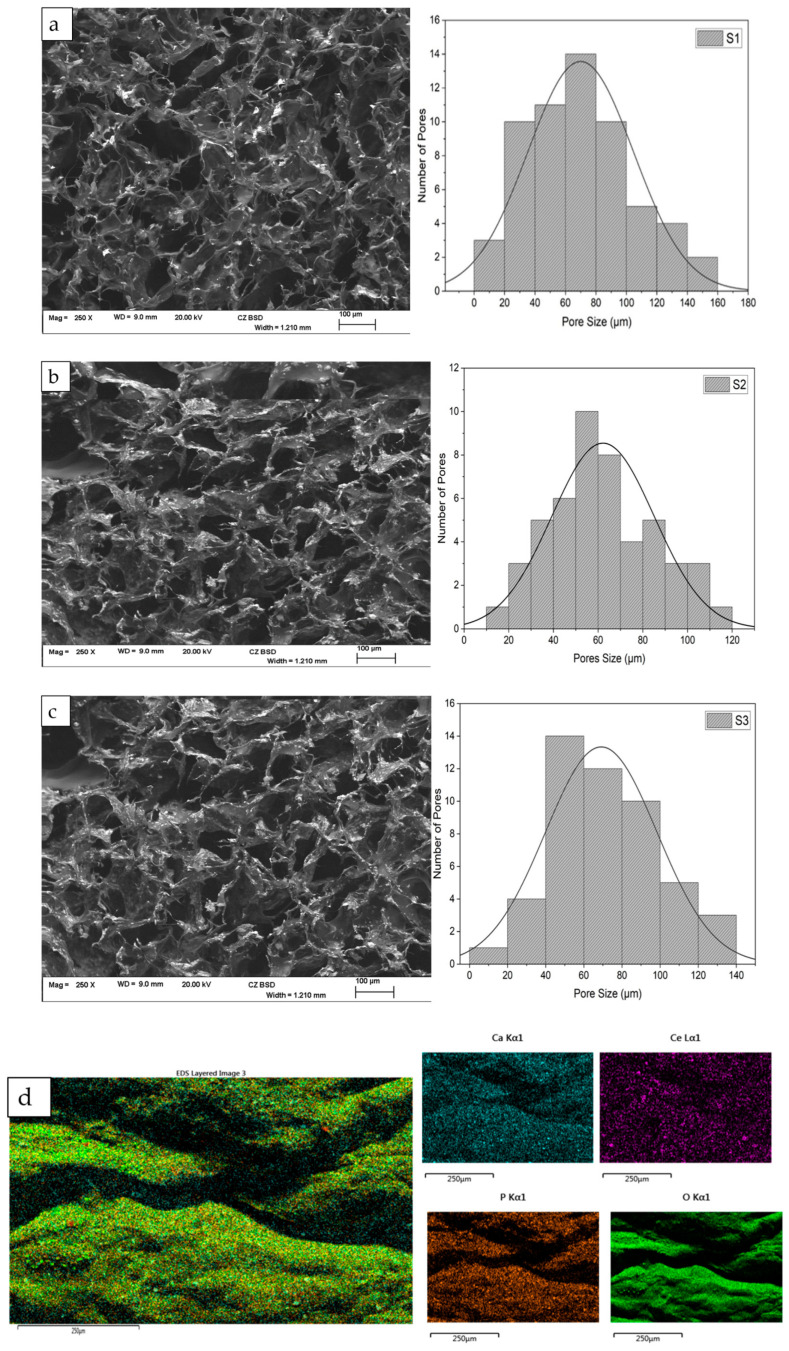
The Hitachi SU8230 SEM, coupled with an energy dispersive X-ray (EDX) detector, was utilised for morphological and elemental characterisation of the freeze-dried chitosan scaffolds (S1, S2, and S3), doped with 10 (mol)% Fe-DCPD mineral and varying concentrations of cerium oxide nanoparticles (S1–10, S2–20, and S3–30 (wt)%): (**a**) S1, (**b**) S2, (**c**) S3, and (**d**) EDX spectrum of the freeze-dried scaffold.

**Figure 4 antibiotics-12-01004-f004:**
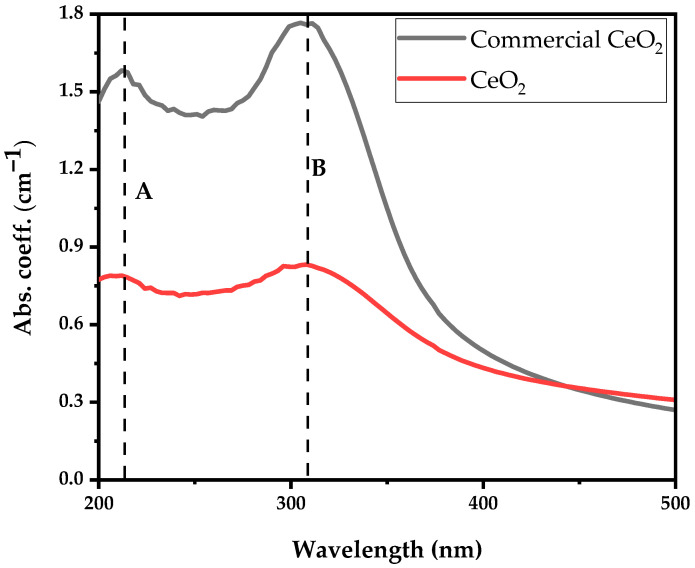
Absorption coefficient spectra of Ce^3+^ and Ce^4+^ ions of commercially available and synthesised CeO_2_ nanoparticles.

**Figure 5 antibiotics-12-01004-f005:**
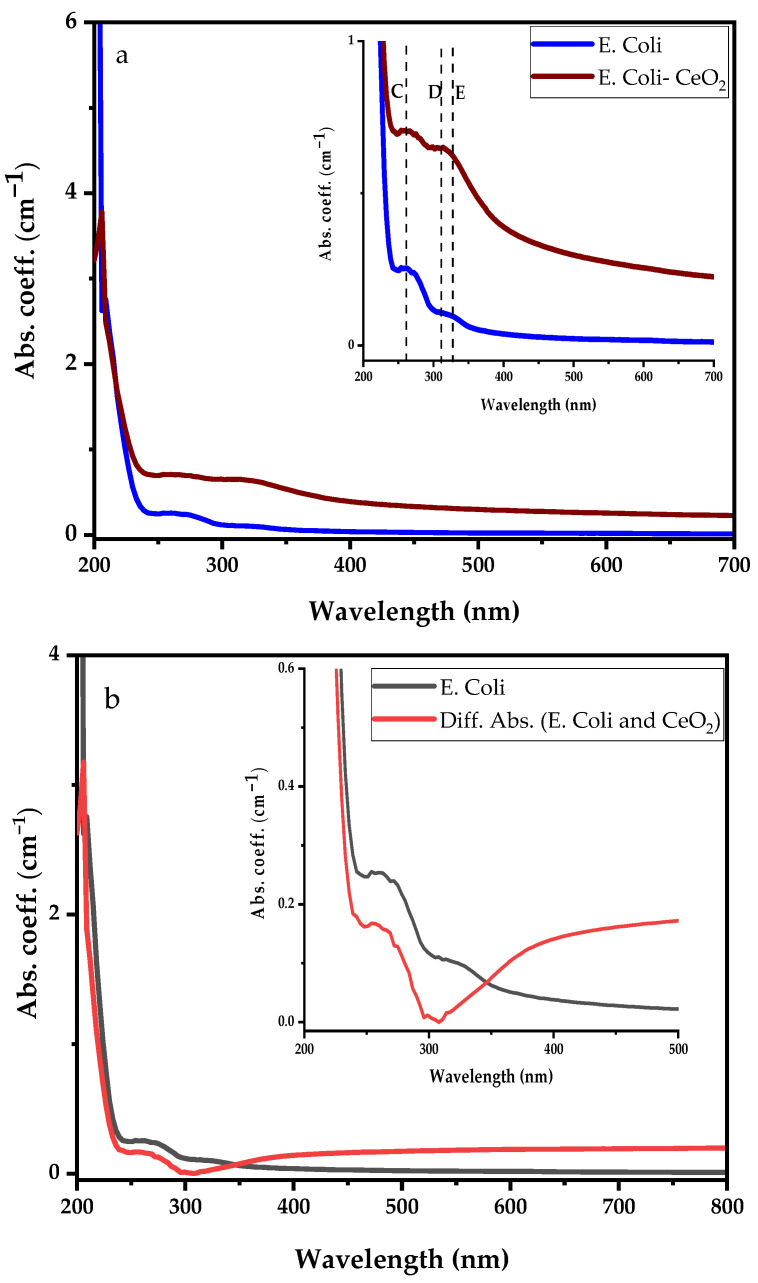
Comparing absorption spectra of *E. coli* with (**a**) *E. coli* dispersed in CeO_2_ nanoparticle suspension solution, and (**b**) differential absorption spectra between CeO_2_ and *E. coli* suspension in CeO_2_ nanoparticle suspension solution.

**Figure 6 antibiotics-12-01004-f006:**
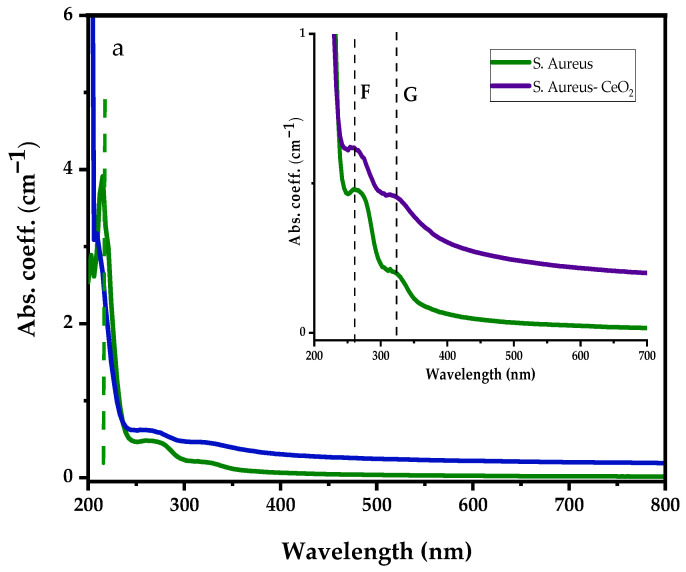
UV-visible absorption spectra of *S. aureus* with (**a**) *S. aureus* dispersed in CeO_2_ nanoparticle suspension solution, and (**b**) differential absorption spectra between CeO_2_ and *S. aureus* suspension in CeO_2_ nanoparticle suspension solution.

**Figure 7 antibiotics-12-01004-f007:**
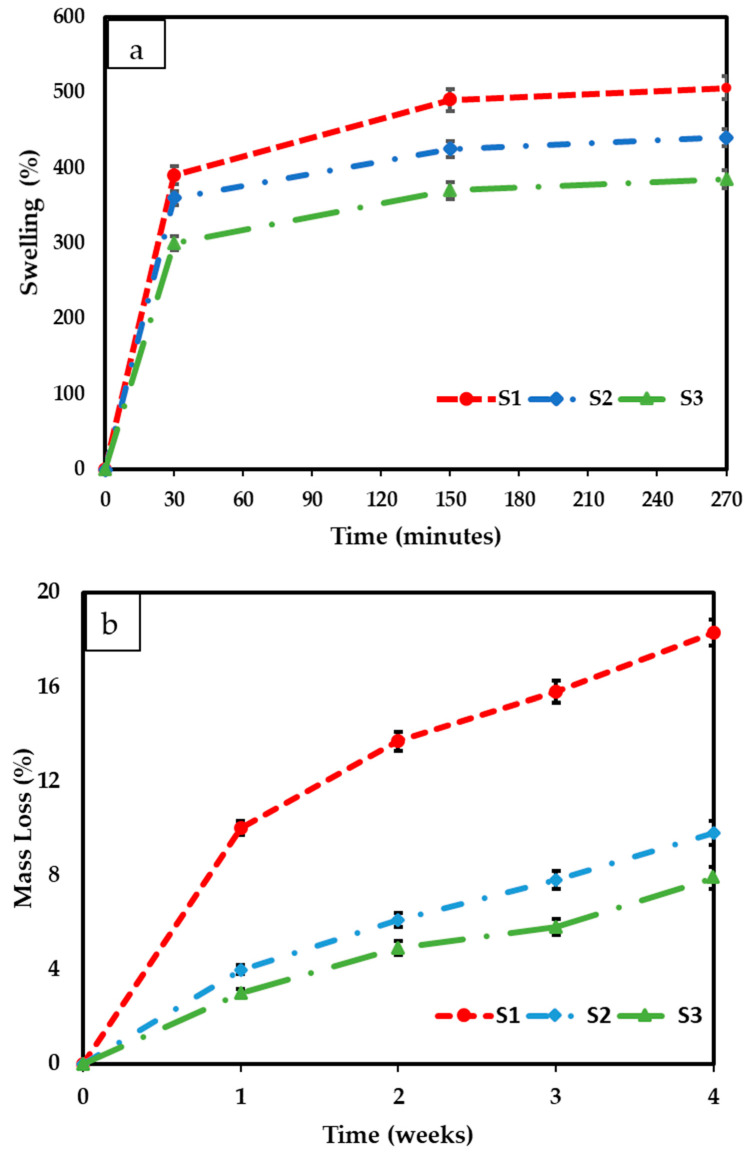
(**a**) The swelling results of different concentrations of cerium oxide freeze-dried scaffolds (10 (wt)%, 20 (wt)%, and 30 (wt)%) and (**b**) the freeze-dried scaffolds’ degradation/mass loss results. All scaffolds were tested in triplicate. The scaffolds were tested in phosphate buffer saline solution at a temperature of 37 °C.

**Figure 8 antibiotics-12-01004-f008:**
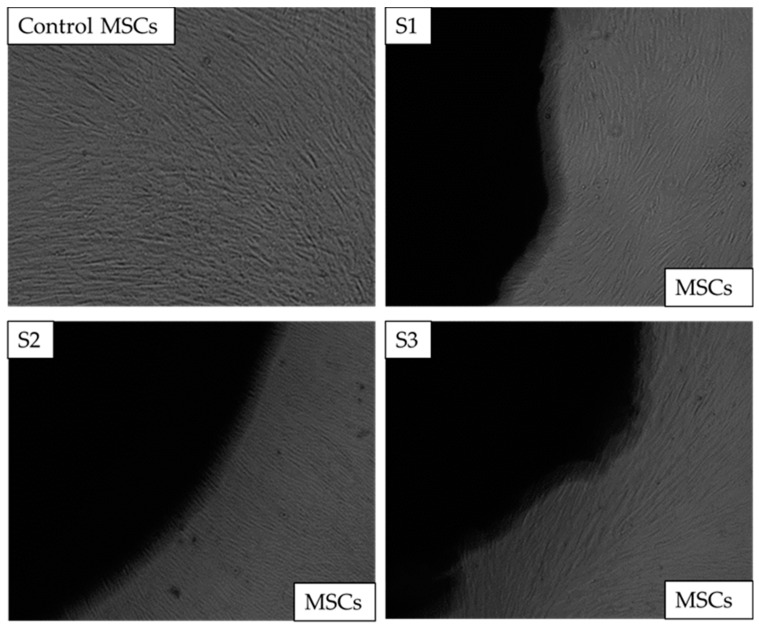
Direct cytotoxicity. The images of chitosan freeze-dried scaffolds containing different concentrations of cerium oxide (10 (wt)% (S1), 20 (wt)% (S2), and 30 (wt)% (S3)) images as captured on the 7th day at x4 The control consisted of bone marrow mesenchymal stem cells (BM-MSCs) without scaffolds.

**Figure 9 antibiotics-12-01004-f009:**
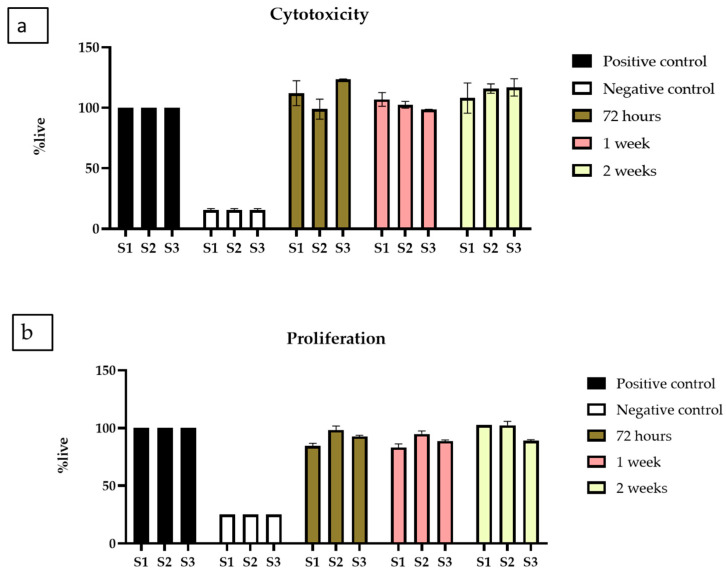
XTT assay (**a**) Cytotoxicity by XTT assay on BM-MSCs when exposed to extracts collected from chitosan freeze-dried scaffolds containing different concentrations of cerium oxide (10 (wt)% (S1), 20 (wt)% (S2), and 30 (wt)% (S3)) chitosan freeze-dried scaffolds. (**b**) Proliferation by XTT assay on BM-MSCs, when exposed to extracts from S1, S2 and S3, presented as the mean and standard error of the mean (SEM).

**Figure 10 antibiotics-12-01004-f010:**
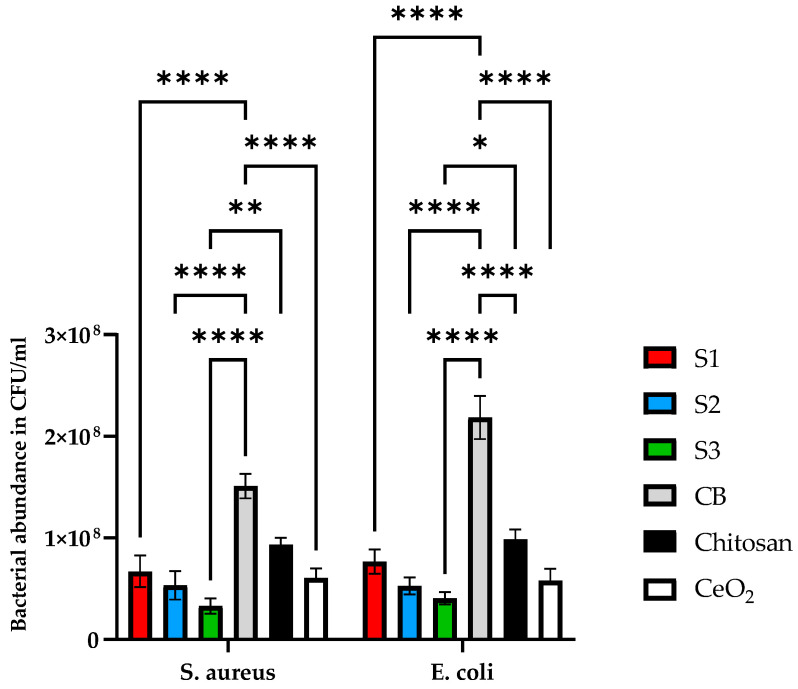
The antibacterial properties of cerium oxide nanoparticles, freeze-dried scaffolds chitosan and (S1, S2, and S3) containing 10, 20, and 30 (wt)% cerium oxide and iron-doped brushite minerals. Antibacterial properties were tested against *S. aureus* and *E. coli.* Bacteria were grown in BHI broth for 24 h and tested using a manual colony counting method. Data are presented as the mean and standard error of the mean (SEM); * *p* < 0.05, ** *p* < 0.01 and **** *p* < 0.0001.

**Table 1 antibiotics-12-01004-t001:** Summary of the Raman peaks assignments for the freeze-dried scaffolds containing iron-doped brushite minerals and cerium oxide nanoparticles.

Wavenumber (cm^−1^)	Assignments	Reference
393	δ(C-C(=O)-C)	[18]
430	PO_4_^3−^	[17]
456	CeO_2_	[19]
590	P-O	[20]
596	δ(C-C=O)	[18]
806	γ(CH)rings	[18]
890	C-C-O	[21]
960	P-O symmetric stretching of PO_4_^3−^	[20]
1046	Fe-CaP	[17]
882–1147	HPO_4_^2−^	[17]
1370–1420	CH_3_	[18,22]
1650	Amide I	[18]
2880	ν(CH)	[18]
2950	CH_3_	[18]

**Table 2 antibiotics-12-01004-t002:** Comparison of physical and biological properties of synthesised scaffolds.

Samples	Pore Size (µm)	Swelling (%)	Degradation (%)	*E. coli*	*S. aureus*	Cytotoxicity
S1	0 to 160	506.9 ± 27.92	18.3 ± 1.76	68 ± 12.00	76 ± 15.00	Non-toxic
S2	10 to 120	440 ± 19.08	9.8 ± 2.08	79 ± 8.00	82 ± 11.00	Non-toxic
S3	0 to 140	385.6 ± 20.3	7.9 ± 1.01	84 ± 7.00	88 ± 6.00	Non-toxic

**Table 3 antibiotics-12-01004-t003:** Summary of the synthesised freeze-dried chitosan scaffold samples embedded with different concentrations of cerium oxide nanoparticles.

Sample	Chitosan (CH)	Iron-Doped Dicalcium Phosphate Dihydrate (10 (mol)% Fe-DCPD)	Cerium Oxide Nanoparticles (CeO_2_)
S1	3 (wt)%	30 (wt)%	10 (wt)%
S2	3 (wt)%	30 (wt)%	20 (wt)%
S3	3 (wt)%	30 (wt)%	30 (wt)%

## Data Availability

The data used to support the findings of this study are available from the corresponding author upon reasonable request.

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
