# Peer review of "Fabrication and Characterisation of the Cytotoxic and Antibacterial Properties of Chitosan-Cerium Oxide Porous Scaffolds"

_antibiotics, 2023, doi:10.3390/antibiotics12061004_

Round 1

Reviewer 1 Report

I consider this manuscript suitable for publication but only after the authors address the following issues.

Major issues: the abstract is to long, it has too much data and details. Moreover, the authors use the word “thus” very often in the first part. Lack of care in writing the manuscript since there is no consistency. It seems that several people wrote different parts but, in the end, did not confirm the structure and coherence of the article.

Images very different from each other in terms of presentation (colors and size). Incorrect identification in some and/or lack of identification in others "a", "b"...

Minor issues:

-          Line 13: repeated sentence.

-          Lines 19: repeated word “healing”.

-          Line 31: or you use the sort versions S. aureus and E. coli, or SA and EC, not both.

-          Line 38: you can put here S. aureus and E. coli or SA and EC, depends on what you choose, but be consistent and always use the same in the entire manuscript.

-          Line 81: move the reference to the end of the sentence.

-          Line 84: what is the meaning of “chitosan in isolation”? Do you mean chitosan per se? Note clear, change to “chitosan per se”.

-          Line 91: put “(HA)” in line 88 after Hydroxyapatite.

-          Line 101: a space is missing before the reference.

-          Line 139: here you use the abbreviation “Fe-DCPD” but in line 24 you used “IB”. Be consistent.

-          Line 145: close the brackets.

-          Line 148: what is the rational of this sentence, it seems out of context.

-          Lines 151 and 152: until here you refer the supplier between parentheses, and now you change to square brackets. Be consistent. It seems like different people wrote different parts of the manuscript.

-          Line 176: “Renishaw in Via” bigger letters size.

-          Line 181: remove the space in the beginning between brackets.

-           

-          Line 206: the open bracket of the supplier is different from the closed one.

-          Line 225: write “in vitro” in italic.

-          Line 227: remove “Phosphate Buffered Saline (DPBS, Life Technologies, Paisley, UK)” and just put DPBS, you already have this abbreviation above in line 206, do not repeat.

-          Lines 236 to 239, line 243 and line 245: you need to be consistent, put the abbreviation between brackets and the supplier between different brackets, as above. And state what DMSO means.

-          Line 241: write “in vitro” in italic.

-          Line 248: repeated word “cells”.

-          Line 261: until here you wrote “hours” and now you write “hrs”. Be consistent.

-          Line 282: FCS or FBS?

-          Line 316: ANOVA not ANOVa.

-          Line 318: write “in vitro” in italic.

-          Line 356: “% of crystallinity”.

-          Line 366: which one is “C”? Identify in the figure 2 A, B and C.

-          Line 370: Figure 2b or 2B? above you wrote 2C, you need to be consistent throughout the text.

-          Line 373: remove the dot before (Figure 2c), and “The percentage of crystallinities”.

-          Line 391: add the reference of this software.

-          Line 399: identify in the figure the letter “d”.

-          Line 410: add a space before the reference.

-          Line 416: “antioxidant and antibacterial agent” instead of “antioxidants and antibacterial agents”; add a space after cerous.

-          Line 417: do you mean “inducing” instead of “induced”?

-          Line 418: add a space before the reference.

-          Lines 414 to 417: long sentence without commas, please rephrase, it is confusing.

-          Line 423: add a space before the reference.

-          Line 431: E. coli in italic.

-          Lines 431, 434 and 441: add a space between E. and coli.

-          Line 443: in Figure 5 you have E.Coli please writ it correctly, you have “a”, “b” and “c” between brackets and in the other figures not. In the caption you have E.coli and some E.coli is written with bigger letters size. Be consistent!

-          Line 451: S. aureus not S.Aureus.

-          Line 464: S. aureus.

-          Line 480: final dot is missing.

-          Line 481: identify the images with a and b.

-          Line 526: Figure 5? Not 10?

-          Line 631: in vivo in italic.

Minor editing of English language required

Author Response

We thank our reviewer for taking the time to read our work and for your valuable feedback. Please see the attached response document.

Reviewer 2 Report

The authors have developed highly porous chitosan scaffolds (S1, S2, and S3) embedded with 30 wt% iron-doped brushite (IB) minerals and varying amounts of cerium oxide nanoparticles (10 wt%, 20 wt%, and 30 wt%). The scaffolds were designed to be biocompatible, porous, and capable of releasing antibacterial agents. This study presents good in vitro testing of antibacterial and materials based research. However, to be accepted into the antibiotics journal, here are some comments to be addressed first. 

Please shorten the abstract text; the length is incorrect.

The introduction is nothing more than a collection of disconnected paragraphs. Please revise.

Figures have no symmetry; there is no A, B, or C clearly mentioned as per the figure captions.

Bacterial results should be antibacterial tests or analyses; also, how slow and fast was the CeO's or agents release for antibacterial findings?

Lines 529–533; Both sentences illustrate the same thing; please delete one and explain why S3 has better antibacterial properties.

The authors stated, "After the completion of day 7, there was no change in the color of the medium, nor was there any turbidity observed, indicating that the cells were healthy." Did authors not refresh the medium for seven days of culture?

The authors have not provided any evidence to support the osteoconductive behavior of the scaffolds. Therefore, It is not appropriate to simply say "Moreover, they exhibited high cell viability in cytotoxicity assays on bone progenitor cells or MSCs, suggesting their potential for osteoconductive properties". It is advised to remove the term "osteoconductive" from the title. 

Please proofread the whole manuscript to eliminate the minor issues for example CAP should be CaP, the full form of DCPD is needed, there should be a space between the word/number and unit, e.g., 3 (%), and remove or correct the full stops of headings and titles.

Author Response

We thank our reviewer for taking the time to read our work and for your valuable feedback. Please see the attachment response document.

Reviewer 3 Report

The paper reports on the preparation of multicomponent scaffolds containing chitosan, cerium oxide and iron-dopred brushite, and the study of their cytotoxicity and antibacterial properties. The subject of the paper fits the scope of Antibiotics journal.

I have the following comments:

1. The choice of the components of the scaffolds is unjustified. What was the purpose of the introduction of iron-containing brushite? Please explain. What is the reason for iron-doping of brushite? Please explain.

2. The existing reports on the biological activity of cerium oxide are largely ignored, while I can suggest that it is the biological activity of CeO2 that dictates the properties of the composite scaffolds. When discussing the properties of CeO2, please refer to the recent papers concerning its antibacterial activity, e.g. doi: 10.1016/B978-0-323-42864-4.00012-9. Similarly, a beneficial effect of cerium oxide on the stem cells has been already reported (see e.g. doi 10.1016/j.msec.2016.05.103). Existing papers on CeO2-containing polymers and biopolymers should be also discussed in the Introduction part of the manuscript.

3. The reaction described by Eq. 1 doesn't lead to CeO2 formation.

4. Please provide the name of the manufacturer of the diffractometer. Please change LAMDA for LAMBDA.

5. XRD pattern cannot be referred to as spectrum.

6. How the crystallinity percentage was determined? Please explain in details. Does pure CeO2 contain amorphous admixture? What is the reason for its formation?

7. Neither SEM nor EDX doesn't allow for determination of phase composition. Please check.

8. It is almost unclear what is the purpose of measuring UV-vis spectra. It is totally unclear how the drop in absorption spectra is related to the decrease in bacterial activity.

9. Please change mesency for mesenchymal.

10. The paper doesn't present resulta of any control experiments. Please provide the comparative data for the composites without brushite and the composites without cerium oxide to elucidate the role of these components.

Moderate editing of English language is strongly recommended

Author Response

We thank our reviewer for taking the time to read our work and for your valuable feedback. Please, see the attached response document.

Round 2

Reviewer 3 Report

The authors have addressed some of my comments. Unfortunately, three comments remain unanswered.

1. The existing reports on the biological activity of cerium oxide are still largely ignored (see my original review report, comment #1). Instead, the authors cited their own paper.

2. The method for determining crystallinity percentage (Lines 200-202) is not generally accepted. I'm not sure that subtracting the area of crystalline peaks from the total area of all peaks in diffractogram would provide any information on the crystallinity of the samples.

3. When discussing UV-vis absorption spectra, the authors mentioned that CeO2 sample exhibit superior antibacterial properties due to high oxidase-like activity. Unfortunately, no measurements of oxidase-like activity were conducted, so this statement is incorrect.

4. No control experiments foir the composites without cerium oxide were conducted. This doesn't allow to elucidate the role of cerium oxide (see my original review report, comment #10).

Minor editing of English language is required

Author Response

We thank our reviewers for their valuable feedback. Please see the attachment of the reviewer's comments response document.

Round 3

Reviewer 3 Report

1. Please provide a correct sodium nitrate formula in Equation 1.

2. In Section 3.1.2 (lines 410, 413), please specify the units (degrees).

3. Diffraction pattern cannot be referred to as spectrum (line 424)

Moderate editing of English language is required

Author Response

We thank our reviewers for their valuable feedback. Please find attached the reviewer's comments response document.
